# FLOWRETRIEVAL: Flow-Guided Data Retrieval for Few-Shot Imitation Learning

**Li-Heng Lin**[†]**, Yuchen Cui**[†]**, Amber Xie**[†]**, Tianyu Hua**[†]**, Dorsa Sadigh**[†*]

**Abstract:** Few-shot imitation learning relies on only a small amount of task-specific demonstrations to efficiently adapt a policy for a given downstream tasks. Retrieval-based methods come with a promise of retrieving relevant past experiences to augment this target data when learning policies. However, existing data retrieval methods fall under two extremes: they either rely on the existence of *exact* behaviors with visually similar scenes in the prior data, which is impractical to assume; or they retrieve based on *semantic similarity* of high-level language descriptions of the task, which might not be that informative about the shared low-level behaviors or motions across tasks that is often a more important factor for retrieving relevant data for policy learning. In this work, we investigate how we can leverage *motion similarity* in the vast amount of cross-task data to improve few-shot imitation learning of the target task. Our key insight is that motion-similar data carries rich information about the effects of actions and object interactions that can be leveraged during few-shot adaptation. We propose FLOWRETRIEVAL, an approach that leverages optical flow representations for both extracting similar motions to target tasks from prior data, and for guiding learning of a policy that can maximally benefit from such data. Our results show FLOWRETRIEVAL significantly outperforms prior methods across simulated and real-world domains, achieving on average 27% higher success rate than the best retrieval-based prior method. In the Pen-in-Cup task with a real Franka Emika robot, FLOWRETRIEVAL achieves 3.7× the performance of the baseline imitation learning techniques that learn from all prior and target data. Website: https://flow-retrieval.github.io

## 1 Introduction

Imitation learning has shown significant promise in robotics with the rise of deep end-to-end models, but a considerable challenge is overcoming its high data demands. Typically, a substantial number of demonstrations, often ranging from hundreds to thousands depending on the complexity and dexterity of the task, are needed to achieve a policy with reliable performance across the expected range of scenarios. This reliance on extensive amounts of data makes it impractical for the policy to adapt quickly to slightly different or out-of-distribution environments or tasks. Therefore, it is crucial to develop methods that facilitate rapid adaptation with small amounts of demonstrations.

One promising approach is to *retrieve relevant data* from collected experiences, which enables more efficient policy learning for a given downstream target task by incorporating valuable information from this prior data. Prior works identify reusable *skills* from pretraining data by training latent representations that capture state-action similarity between the target and prior data [1, 2]. However, these methods often assume that the exact visual observation present in the target task exists in the robot's prior experiences, and thus primarily retrieve based on such visual similarities. In practice, this assumption is rarely met in real-world applications. For instance, the task of "rotating a doorknob" appears to be visually different from "rotating a faucet". Nevertheless, the movement involved in one task ought to be applicable and beneficial for the other, regardless of their visual disparities.

Our key insight is that prior datasets can serve a broader purpose than merely retrieving the same skills of visually similar states. Target task data may in fact exhibit similarities to prior data in

---

* † Computer Science Department, Stanford University, Stanford, CA, USA.

8th Conference on Robot Learning (CoRL 2024), Munich, Germany.

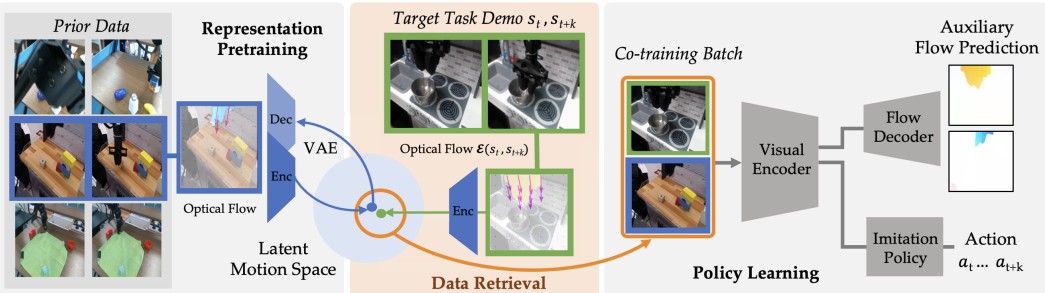

Figure 1: **Overview of FLOWRETRIEVAL.** We first learn a motion-centric latent space through training a VAE for embedding optical flows; then retrieve prior data similar to target data by measuring pairwise distances in the learned latent space; and we augment policy co-training with a supervised prediction loss for optical flow.

terms of low-level *motion*, offering an opportunity for knowledge transfer of motions. For example, both turning a doorknob or turning a faucet necessitate a similar rotating behavior, despite task dissimilarity. We posit that retrieving the robot's past experiences with similar motion patterns to the downstream task are particularly valuable for learning. The idea of retrieving and using data with similar *behaviors* is not new. Recent works often leverage language descriptions of tasks for retrieving similar behaviors [3–5]; however, language often operates at a much higher level of abstraction and cannot readily capture low-level motions such as turns and twists that could in fact be beneficial and transferable across prior and target tasks. Building upon this insight, we introduce FLOWRETRIEVAL, an approach for retrieving and learning from motion-similar data from prior datasets.

We illustrate an overview of FLOWRETRIEVAL in Fig. 1. Our approach leverages *optical flow* as a representation for motion-similarity based retrieval. FLOWRETRIEVAL first acquires a motion-centric latent space by computing optical flow between the current frame and a future frame of the robot's RGB visual observations, and employing a variational autoencoder (VAE) to embed the optical flow data. During the data retrieval process, we rank prior data points based on their distance to target task data in the latent space and retrieve the closest ones. We then train the imitation policy network using an augmented dataset through co-training with the target-specific and retrieved data. During policy learning, FLOWRETRIEVAL leverages an auxiliary loss of predicting the optical flow as additional guidance for representation learning, encouraging the model to encode the image with enough details to predict optical flow alongside predicting the action.

We demonstrate FLOWRETRIEVAL achieves higher performance than prior approaches both in simulation and real robot environments. We experiment with two simulation tasks: Square Assembly task with a mixture of similar and adversarial tasks as prior data [2], and a pick-and-place Can task from the LIBERO benchmark, with a random subset of the LIBERO tasks as prior data [6]. Our real robot experiments spans two settings with pick-and-place tasks as the target task: i) using a Widow-X robot with the Bridge [7] dataset as the prior data, and ii) using a Franka Panda robot with either past data from other similar tasks on the same robot or sub-sampled DROID [8] dataset as the prior data. Our experiments demonstrate FLOWRETRIEVAL achieves an average of 14% higher success rate than the best baseline in each domain (+10% in simulation,+19% in real). FLOWRETRIEVAL also achieves on average 27% higher success rate than the best prior retrieval method.

## 2   Problem Statement

We model our *imitation learning* problem via a Markov decision process (MDP), where an agent moves between states $s \in \mathcal{S}$ by taking actions $a \in \mathcal{A}$. The transition between states is governed by the dynamics function $\mathcal{P} : \mathcal{S} \times \mathcal{A} \to \mathcal{S}$. An underlying reward function $\mathcal{R} : \mathcal{S} \to \mathbb{R}$ defines achieving the target task but is unobserved by the learning agent. A policy $\pi(a|s)$ is the conditional distribution of actions at each state. We assume the human demonstrator's policy $\pi_E$ is near-optimal under the target task reward function. The goal of imitation learning is to find a policy $\hat{\pi}(a|s)$ that imitates

expert behavior $\pi_E$ with dataset $\mathcal{D}_\text{target} = \{(s,a)|a \sim \pi_E(a|s,r)\}$ containing demonstrations of the target task, where $r$ denotes the reward of achieving the target task.

In our *few-shot imitation* setting, $\mathcal{D}_\text{target}$ is assumed to contain only a few demonstrations—not enough to be able to learn the task from scratch. We additionally assume access to a prior dataset $\mathcal{D}_\text{prior} = \{(s,a)|a \sim \pi_E(a|s,\cdot)\}$ that consist of data of the robot performing diverse tasks. The goal of data retrieval is to extract the most relevant subset of prior data ($\delta\%$ of $\mathcal{D}_\text{prior}$) to augment $\mathcal{D}_\text{target}$ and improve the learning of $\hat{\pi}$.

In this work, we decouple the problem of few-shot imitation learning via data retrieval into two sub problems: 1) the **data retrieval** problem focuses on constructing a similarity function $\mathcal{S}$ to extract the desired data from prior dataset (*"what to retrieve?"*), and 2) the **policy learning** problem is concerned with how to maximally utilize the retrieved data $\mathcal{D}_\text{retrieved}$ for improving performance on the target task (*"how to use retrieved data?"*). Decoupling the two problems allows us to independently evaluate the performance of data retrieval irrespective of the choice of policy learning, and further evaluate policy learning performance under the assumption of having access to the most relevant and valuable retrieved data. At the core of the data retrieval problem is to define a measure of similarity $\mathcal{S}(p,t)$ that compares data points $(s_p, a_p) \in \mathcal{D}_\text{prior}$ to $(s_t, a_t) \in \mathcal{D}_\text{target}$. The top $\delta\%$ of $\mathcal{D}_\text{prior}$, ranked by $\mathcal{S}$, will be retrieved. With the retrieved data, the problem of policy learning aims to improve performance on target task by maximally capitalizing on the common patterns and regularities in the retrieved data. Given $\mathcal{D}_\text{retrieved}$ and $\mathcal{D}_\text{target}$, the goal is to find a policy $\hat{\pi}$ that can outperform learning only from $\mathcal{D}_\text{target}$ or even learning from naïvely combining $\mathcal{D}_\text{prior}$ and $\mathcal{D}_\text{target}$.

## 3   Related Work

Learning from demonstrations in a data-efficient way has been a long-sought goal of robot learning. The full literature is beyond the scope of this work. Our work is closely related to the body of work that leverages prior data for few-shot imitation of visuomotor policy. *Pretraining and finetuning* is the most common scheme of few-shot imitation learning that leverages prior datasets [9–11]. However, this two-stage scheme often faces challenges in balancing the extent of freezing and tuning parameters to achieve optimal performance. *Co-training* has emerged as a popular approach by directly incorporating both prior and target data in the same training batch [1, 2, 12, 13]–albeit requiring a balancing act between prior and target data –achieving competent performance when target data is scarce. In this work, we employ the co-training scheme for training the downstream policy network. Additionally, the choice of prior data in the training batch can significantly affect policy performance [11, 14]. Our approach, FLOWRETRIEVAL carefully filters the prior dataset based on measuring motion similarity of the provided demonstrations from the target task.

Extracting task-relevant data from prior datasets can be achieved with different similarity metrics—from high-level language descriptions to dense visual dynamics (Fig. 2). In the rest of this section, we examine these different types of motion guidance and discuss how they are used to solve the two sub problems in our problem setting as discussed in Section 2.

| Guidance Type | | Language | Feature Traces | Optical Flow | Visual Dynamics |
|---|---|---|---|---|---|
| **Example** | | *"Pick up the nut"* 

 *"Insert into square peg"* |  |  |  |
| **Attributes** | Granularity | Coarse | Medium | Medium | Fine |
| | Label Source | Human Annotation | Off-the-Shelf Vision Model | Off-the-Shelf Vision Model | Ground-Truth Video |
| | Information | Semantic | Motion | Shape + Motion | Texture + Shape + Motion |

Figure 2: **Different types of motion guidance for imitation learning.** Here, we present a spectrum of different types of motion guidance used in the literature for either retrieval or directly learning the policy. This spectrum varies the level of granularity of motion starting from coarse semantic information via language from the left to guidance about feature traces or optical flow capturing motion and shape of objects all the way to more fine-grained guidance such as visual dynamics that also capture the texture of the scene.

**Language Descriptions.** On one end of the spectrum in Fig. 2, we can retrieve base on high-level language descriptions to reuse semantically similar prior experiences. Recent work has explored using text to retrieve similar past experience for novel target tasks [3, 5, 15]. At the same time, a body of work leverages language for improving policy learning through explicit representation learning [16–22] or using language prediction as an auxiliary task [3, 23–25]. However, we argue that language descriptions may not capture the detailed information for retrieving the most relevant data or guiding policy learning to properly reuse prior data. For example, the same description of "open the door" can apply to opening doors with either a knob or a handle, each requiring a different low-level motion – hence not all semantically similar motions are useful for the target task. At the same time, the rotating motion of "turn on the faucet" might also be useful for learning how to turn a door knob, and focusing on semantic similarity will oversee such data points. In this work, we focus on leveraging fine-grained low-level similarity metrics that can capture motion similarity.

**Visual Dynamics.** On the other end of the spectrum in Fig. 2, we can consider similarity based on visual dynamics, which compares tasks at the pixel level. *Behavior Retrieval* [2] pretrains a variational auto-encoder over state-action pairs and uses similarity in that latent space for filtering prior data. Similarly, *SAILOR* [1] focuses on identifying reusable skills from prior data by pretraining a latent space via an inverse dynamics loss on actions with sequences of images as input. Recent work of Palo and Johns [26] directly compares extracted visual features at annotated bottleneck states for retrieving and replaying recorded trajectories. However, in all of these works, directly embedding RGB observations couples visual similarity with task similarity and hinders the latent space's ability to extract data from visually different tasks that are similar at the motion level.

**Flow-based Motion Guidance.** There are intermediate motion representations that do not suffer from the shortcomings of language or visual dynamics similarity. Feature traces and optical flow (shown in the middle range of Fig. 2) are some examples that consider motion similarity at an intermediate level. However, most prior work often use them directly within the policy learning paradigm and not as a retrieval method [27–31]. FLOWRETRIEVAL, instead attempts to use these intermediate representations for retrieval enabling a more policy-agnostic approach when tapping into prior data. Additionally, FLOWRETRIEVAL leverages an auxiliary reconstruction loss on optical flow for policy learning to encourage the model to learn motion-centric representations. Such guidance as an auxiliary task allows FLOWRETRIEVAL to be agnostic to the backbone imitation learning algorithm and therefore FLOWRETRIEVAL can be applied to different existing policy learning models unlike prior work that use intermediate motion representations.

## 4  FLOWRETRIEVAL: Flow-Guided Data Retrieval for Imitation Learning

FLOWRETRIEVAL consists of three stages: 1) motion representation pretraining, 2) data retrieval, and 3) policy learning. We discuss each in details below.

### 4.1  Motion-Centric Pretraining

To retrieve data with similar motion patterns, it is essential to have a latent space encoding observations based on their motion similarity. In particular, existing robot datasets, which contains similar motion patterns, often are collected in diverse environments requiring the motion-centric representation to exclude irrelevant features such as background information from the images. Optical flow presents an ideal solution as it encodes the motion of pixels without being influenced by other features such as object textures. However, optical flow data is inherently high-dimensional, and direct distances between raw optical flows isn't meaningful. Hence, we employ a straightforward variational autoencoder (VAE) to embed optical flow information, creating a latent space that facilitates the assessment of similarity between optical flows.

We process the entire $\mathcal{D}_{\text{prior}}$ with flow operator $\mathcal{E}_{\text{flow}}$ to find optical flow $f_t$ for each state $s_t$ in $\mathcal{D}_{\text{prior}}$. In our experiments, we leverage GMFlow [32] for computing the optical flow between visual state $s_t$ and a future frame $s_{t+k}$. The value of $k$ is determined by downstream learning policy's action prediction length. For standard behavioral cloning (BC) algorithms that predict a single action as output, we set $k = 1$. For models that predict a sequence of actions such as diffusion policy [33, 34],

we set $k$ to match the action length the model predicts at each step ($k = 16$ in our experiments). We denote the matching ordered set of optical flow for $\mathcal{D}_{\text{prior}}$ as $\mathcal{F}_{\text{prior}}$:

$$\mathcal{F}_{\text{prior}} = \{\mathcal{E}_{\text{flow}}(s_t, s_{t+k})|s_t, s_{t+k} \in \mathcal{D}_{\text{prior}}\} \tag{1}$$

The VAE consists of an encoder $p_\theta(f_t)$ and decoder $q_\phi(\cdot)$, and is trained with a reconstruction loss:

$$\mathcal{L}_{\text{VAE}}(\theta, \psi) = \mathbb{E}_{f_t \sim \mathcal{F}_{\text{prior}}}||q_\psi(p_\theta(f_t)) - f_t||_2 \tag{2}$$

If $\mathcal{D}_{\text{prior}}$ is large enough, we expect the VAE trained only on $\mathcal{F}_{\text{prior}}$ to generalize to the target task data $\mathcal{F}_{\text{target}}$, as the domain difference in optical flow is much smaller than that in the RGB space.

## 4.2 Data Retrieval with the Learned Motion-Centric Latent Space

The data retrieval process involves assessing every data point in the prior dataset based on its relative similarity to the target task data, subsequently extracting the highest-ranked portion of these similar data points. Given $D_{\text{target}}$, we process every state to find the optical flow the same way as we process $\mathcal{D}_{\text{prior}}$. We then compute the similarity score for each state in prior data by computing pairwise distances to target task data and taking the minimum value. The similarity function is:

$$\mathcal{S}(s_i) = - \min_{\forall f_j \in \mathcal{F}_{\text{target}}} ||p_\theta(f_i) - p_\theta(f_j)||_2 \, , f_i = \mathcal{F}_{\text{prior}}[i] \tag{3}$$

We select the top $\delta\%$ of the prior dataset ordered by the similarity score. Specifically, the similarity threshold can be found at the $\lceil \delta N \rceil$-th element of the sorted dataset, where $N = ||\mathcal{D}_{\text{prior}}||$:

$$\eta = \text{sorted}(\mathcal{S}(s_i)|s_i \in \mathcal{D}_{\text{prior}})[\lceil \delta N \rceil] \tag{4}$$

The retrieved dataset thereby contains all data with similarity score higher than $\eta$:

$$\mathcal{D}_{\text{retrieved}} = \{(s, a)_{t:t+k}|(s, a)_{t:t+k} \in \mathcal{D}_{\text{prior}} \text{ and } \mathcal{S}(s_t) > \eta\} \tag{5}$$

In practice, it may not be feasible or necessary to scan through all prior data. One can randomly sample a large enough candidate set of data to serve as $\mathcal{D}_{\text{prior}}$ as suggested by Nasiriany et al. [1].

## 4.3 Flow-Guided Learning

To train the downstream policy, it is not sufficient to only include the retrieved data with similar motion. To ensure the policy pays attention to the retrieved motion, we train the policy with both an action prediction loss and an auxiliary loss for predicting the optical flow. In contrast to approaches that require the model to predict dense visual guidance as a bottleneck layer before the action head or directly reconstruct the action from optical flow [27, 29], FLOWRETRIEVAL treats optical flow prediction only as an auxiliary task. This is due to the fact that requiring the model to predict such dense visual guidance leads to encoding unnecessary details for predicting end-effector motion. Thus, we employ optical flow prediction as an auxiliary task to provide dense guidance for policy learning without imposing the challenge of learning a significantly more complex task.

FLOWRETRIEVAL is agnostic to the choice of imitation learning algorithm for policy learning. In our experiments, we implemented FLOWRETRIEVAL with diffusion-based policy [7, 33]. We assume the BC backbone has some visual encoder $\phi_{\text{enc}}$ that embeds visual inputs into a vector form and a policy head $\phi_{BC}$ that predicts the action output. The visual encoder can optionally start from a pretrained representation (e.g. ImageNet-pretrained weights). We incorporate an additional flow decoder ($\phi_{\text{aux}}$) from the bottleneck representation layer to enforce the auxiliary task loss. For co-training, we sample equal amounts of data from $\mathcal{D}_{\text{retrieved}}$ and $\mathcal{D}_{\text{target}}$ into each batch $\mathcal{B}$. Concretely, the loss for FLOWRETRIEVAL policy learning is:

$$\mathcal{L}(\phi_{\text{enc}}, \phi_{BC}, \phi_{\text{aux}}) = \mathbb{E}_{(s,a)_{i:i+k} \in \mathcal{B}}[\mathcal{L}_{BC}(s_i, a_{i:i+k}) + \lambda||\phi_{\text{aux}}(\phi_{\text{enc}}(s_i)) - \mathcal{E}_{\text{flow}}(s_i, s_{i+k})||_2] \tag{6}$$

where $\lambda$ is a hyperparameter (we use $\lambda = 0.01$ in all of our experiments), and $\mathcal{L}_{BC}$ is a supervised $L_2$ loss on action prediction. Algorithm 1 presents the pseudo code of FLOWRETRIEVAL: line 1-3 pretrain the representations, line 4-8 retrieve the relevant data, and line 9-13 train a policy using both the target and retrieved data.

---

**Algorithm 1** FLOWRETRIEVAL ($\mathcal{D}_{\text{target}}$, $\mathcal{D}_{\text{prior}}$, $k$, $\delta$, $\mathcal{E}_{\text{flow}}$)

---

**Require:** demos $\mathcal{D}_{\text{target}}$, prior data $\mathcal{D}_{\text{prior}}$, action chunking length $k$, retrieval threshold $\delta$, flow model $\mathcal{E}_{\text{flow}}$;
1: /* 1. Representation Pretraining (can be done only once for fixed prior data) */
2: $\mathcal{F}_{\text{prior}} = \{\mathcal{E}_{\text{flow}}(s_t, s_{t+k}) | s_t \in \mathcal{D}_{\text{prior}}\}$;                    ▷ compute optical flow for prior data
3: update $\theta, \psi$ by minimizing $\mathcal{L}_{\text{VAE}}(\theta, \psi) = \mathbb{E}_{f_t \sim \mathcal{F}_{\text{prior}}} ||q_\psi(p_\theta(f_t)) - f_t||_2$ ;          ▷ train VAE with Eq. (2)
4: /* 2. Data Retrieval */
5: $\mathcal{F}_{\text{target}} = \{\mathcal{E}_{\text{flow}}(s_t, s_{t+k}) | s_t \in \mathcal{D}_{\text{target}}\}$;                ▷ compute optical flow for target data
6: $\bar{\mathcal{S}} = (- \min_{f_j \in \mathcal{F}_{\text{target}}} ||p_\theta(f_i) - p_\theta(f_j)||_2 | f_i \in \mathcal{F}_{\text{prior}})$;          ▷ compute similarity scores with Eq. (3)
7: $\eta = \text{sorted}(\bar{\mathcal{S}})[\lceil \delta ||\mathcal{D}_{\text{prior}}|| \rceil]$;                                    ▷ find retrieval threshold Eq. (4)
8: $\mathcal{D}_{\text{retrieved}} = \{(s,a)_{t:t+k} | (s,a)_{t:t+k} \in \mathcal{D}_{\text{prior}} \text{ and } \bar{\mathcal{S}}[t] > \eta\}$                    ▷ retrieve data Eq. (5)
9: /* 3. Policy Learning */
10: randomly initialize policy model $\phi_{\text{enc}}, \phi_{BC}, \phi_{\text{aux}}$ (optionally load pretrained weights for $\phi_{\text{enc}}$)
11: **repeat**
12:     sample batch $\mathcal{B}_{0:b} \sim \mathcal{D}_{\text{target}}$, $\mathcal{B}_{b+1:2b} \sim \mathcal{D}_{\text{retrieved}}$ to update $\pi$ with loss $\mathcal{L}(\phi_{\text{enc}}, \phi_{BC}, \phi_{\text{aux}})$;      ▷ Eq. (6)
13: **until** $\pi$ converged; **return** $\pi$

---

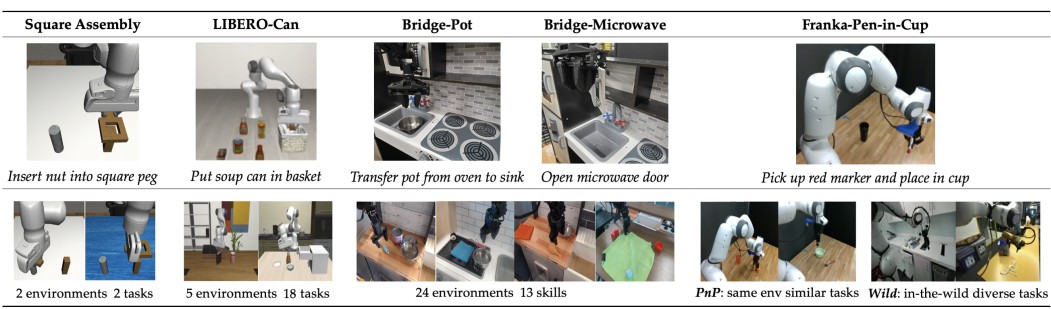

Figure 3: **Experimental Setup.** We experiment with 5 different manipulation tasks for evaluating FLOWRE-TRIEVAL. Bottom row shows the prior dataset we used in each experiment and their meta data.

## 5 Experiments

In this section, we first introduce the task domains and the baselines we compare with, and then present the quantitative results of policy learning and qualitative analysis of retrieval.

**Task Domains.** We evaluate FLOWRETRIEVAL on table-top manipulation tasks of varying difficulty and different compositions of prior datasets (a total of 5 tasks shown in Fig. 3):

- **Square Assembly** is a peg insertion task from the Robomimic benchmark [35]. Behavior Retrieval [2] adapted this task to evaluate data retrieval. $\mathcal{D}_{\text{target}}$ consists of 10 demonstrations. $\mathcal{D}_{\text{prior}}$ consists of two different types of data: 200 episodes of *useful* demonstrations placing the nut into the goal peg, and 200 episodes of *adversarial* demonstrations that place the nut into the wrong peg. We further adapted this task to emulate an extreme case where the target task is in the same environment as the adversarial data while the useful data has a different background.

- **LIBERO-Can** is a pick-and-place task from the LIBERO benchmark [6]. $\mathcal{D}_{\text{target}}$ consists of 10 human demonstrations from the LIBERO benchmark, and $\mathcal{D}_{\text{prior}}$ consists of 18 selected tasks from LIBERO-90 and each with 50 trajectories, resulting in a total of 900 trajectories. Half of the prior data are pick-and-place motions that are intuitively useful for learning the target task, and the other half are tasks involving other diverse motions such as opening a drawer.

- **Bridge-Pot** and **Bridge-Microwave** are two real robot tasks in a toy kitchen setting. We leverage Bridge-V2 [7] as $\mathcal{D}_{\text{prior}}$ and collected $\mathcal{D}_{\text{target}}$ in our own setup with a ViperX arm [36], which is similar to some environments in Bridge-V2 but does not exist in the prior dataset. We collected 10 demonstration for each of the target tasks and use a subset of Bridge-V2 as prior data.

- **Franka-Pen-in-Cup** is a real task with the Franka Emika Panda robot arm. $\mathcal{D}_{\text{target}}$ consists of 10 human demonstrations. We tested two instances of $\mathcal{D}_{\text{prior}}$: 1) *PnP (Pick-and-Place)*: 105 trajectories of pick-and-place tasks from the same robot, and 2) *Wild*: 400 randomly sampled trajectories from the DROID [8] dataset (filtered to roughly match the viewpoint in target task). *PnP* contains semantically similar data to target task while *Wild* contains a diverse range of tasks.

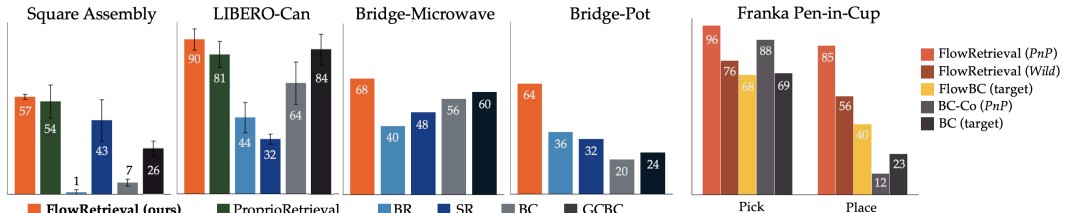

Figure 4: **Quantitative Results.** We plot success rates (%) of learned policies and observe that FLOWRETRIEVAL outperforms baselines across tasks. Simulation results are averaged over 2 training seeds and 3 evaluation seeds (50 rollouts each). Real results are from 25 rollouts of best-of-last-3 checkpoints.

**Baselines.** We leverage co-training for downstream policy learning for all baselines that use prior data, and each training batch contains a mixture of half target task data and half prior data. In the simulation domains and real-world Bridge tasks, we compare FLOWRETRIEVAL with several baseline imitation learning algorithms including prior data retrieval methods (full details in Appendix):
- **BC** is standard behavior cloning with diffusion policy [33] trained only on target data;
- **GCBC** is goal-conditioned diffusion policy trained on all data;
- **BR** (Behavior Retrieval) is a prior work [2] that trains a VAE for state-action pairs for retrieval;
- **SR** (SAILOR Retrieval) is the pretraining algorithm from SAILOR [1]; we use the pretrained latent skill space for retrieving data but keep policy learning consistent with other methods;
- **ProprioRetrieval** is a naïve method where we use proprioceptive data for computing similarity.

In Franka-Pen-in-Cup task, we focus on evaluating FLOWRETRIEVAL with different types of prior dataset (*PnP* and *Wild* in Fig. 3) and compare it with a co-training-only baseline **BC-Co** – co-trained on all of prior data and target data – as well as an ablation **FlowBC** that applies flow loss during policy learning from target task data without retrieval.

**Quantitative Results.** Fig. 4 presents the success rates of different methods for each task. FLOWRETRIEVAL outperforms the best baseline method across different tasks, achieving an average of 14% higher success rate than the best baseline method in each domain (+10% in simulation,+19% in real). FLOWRETRIEVAL also achieves on average 27% higher success rate than the best retrieval-based prior method. We observe that **BR** and **SR** (shown in blue) can perform worse than **BC** (shown in grey), which is potentially due to retrieving harmful data from the prior dataset. **ProprioRetrieval** (shown in green) performs competently in the two simulated tasks. However, it can fail in more complex settings (e.g. when camera viewpoints in prior data vary), due to the fact that it is ignorant of the visual state (more in Appendix). In the real *Pen-in-Cup* task with a Franka Emika robot (right of Fig. 4), FLOWRETRIEVAL (shown in orange) achieves 3.7× the performance of the imitation baseline, learning from all prior and target data (shown in grey).

**Ablation of Data Composition.** Fig. 4 (right) shows the evaluation and ablation of FLOWRETRIEVAL with different dataset compositions in *Franka Pen-in-Cup* task. In this task, the *pick* stage of the target task shares similarity with a prior dataset (*PnP*, which consists of mostly pick and place tasks) while the *place* stage of the task employs a unique strategy that is very different from this prior dataset. We see that **BC-Co** (in grey) with *PnP* improves *pick* performance compared to BC (in black) but performs worse for the overall pick and place task. FLOWRETRIEVAL (in orange) instead successfully filters out the harmful data in *PnP* and achieves the highest performance. FLOWRETRIEVAL (in dark orange) also achieves performance higher than **FlowBC** (+16%) (in yellow) in this task by retrieving from in-the-wild prior data. Note that due to the large discrepancy between viewpoints in prior and target data, we do not use the action retrieved from *Wild* for action prediction.

**Qualitative Analysis.** Fig. 5 visualizes example retrieved data points by each method[2]. FLOWRETRIEVAL retrieves similar motion from prior data while **BR** and **SR** retrieve based on visual similarity of the state and consequently can retrieve adversarial data. In the *Square Assembly* example, both baselines end up retrieving data points where the robot moves towards the round peg. FLOWRETRIEVAL on the other hand selects a very similar motion toward the square peg even when the

---

[2]Additional visualization of retrieved data for each task can be found in Appendix and on project website.

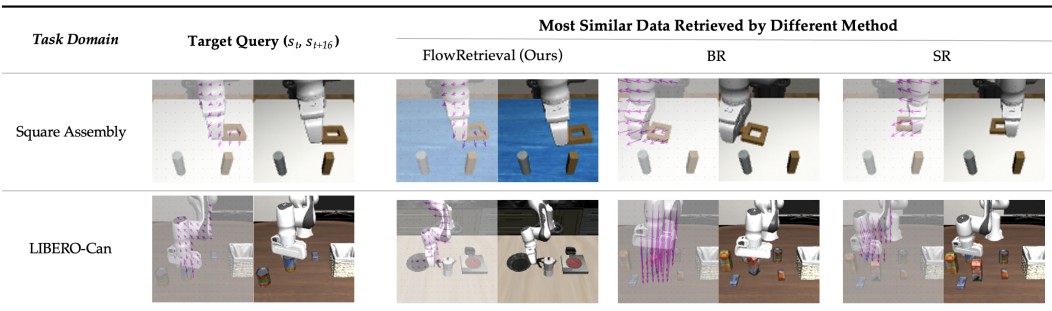

| *Task Domain* | **Target Query** ($s_t, s_{t+16}$) | **Most Similar Data Retrieved by Different Method** | | |
|---|---|---|---|---|
| | | FlowRetrieval (Ours) | BR | SR |
| Square Assembly | | | | |
| LIBERO-Can | | | | |

Figure 5: **Qualitative Analysis.** We visualize most similar motion in prior dataset to queries from target task demonstrations. For each example data point, we show the image observations $s_t$ and $s_{t+16}$, then overlay the optical flow on top of $s_t$. We see that FLOWRETRIEVAL focuses on retrieving similar motions to target queries while baselines may retrieve visually similar states with different motions.

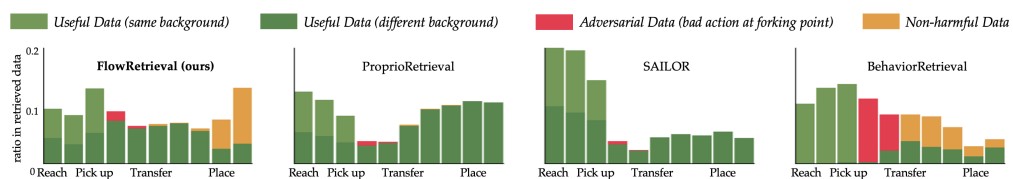

Figure 6: **Retrieval Distribution in Square Assembly.** We visualize the distribution and composition of retrieved data by different retrieval methods. FLOWRETRIEVAL mostly retrieves useful data (green) for each stage of the task – especially during the challenging parts of picking up and transferring the square to the correct peg – and filters out the majority of adversarial data (red).

background color is completely different from that in target data. In *LIBERO-Can*, **BR** and **SR** again focus mainly on visual similarity instead of the picking up motion and retrieved data points that have very different motion (moving downwards). However, FLOWRETRIEVAL retrieves the correct motion of picking up, even if it is picking up a different object than the one in the target task.

Additionally, we consider the Square Assembly task, where we have a curated prior dataset. Specifically, we know the data right after the robot picks up the nut is crucial: we label the actions that move towards the wrong goal after picking up as *adversarial* data. We label the data points after the forking point for transferring the square on the decided peg as *non-harmful*. We can therefore analyze the quality of retrieved data under different retrieval methods in the *Square Assembly* task by plotting the amount of different types of data retrieved from each stage of the task (split into 10 bins) as shown in Fig. 6. FLOWRETRIEVAL uniformly retrieves from prior useful data and retrieves little adversarial data, while baseline models either cannot effectively filter out the adversarial data or do not retrieve enough useful data at the bottleneck stage (between *pick up* and *transfer*) of the task.

## 6 Conclusion

**Summary.** We propose FLOWRETRIEVAL, a method for few-shot imitation learning that retrieves motion-similar data from heterogeneous prior datasets via optical flow. FLOWRETRIEVAL addresses the limiting assumption of prior data retrieval methods that often require visually similar experiences to exist in prior dataset. In our experiments, FLOWRETRIEVAL achieves significantly better performance (+14% on average) than other retrieval-based methods.

**Limitations and Future Work.** Pretraining with large-scale datasets requires extensive computation but is done only once. In contrast, FLOWRETRIEVAL processes prior data for each new target task, calculating and sorting pair-wise distances, which can add overhead and doesn't scale well with large datasets. Caching embeddings and sub-sampling can help, but practical deployment remains challenging, a limitation left for future work. Meanwhile, the optimal retrieval threshold is task-dependent and needs manual tuning for best performance. Future work could automate threshold discovery using annotated boundary data or active learning methods.

## Acknowledgments

This work is supported by NSF projects #1941722, #2125511, #2218760, ONR project N00014-21-1-2298, DARPA #W911NF2210214. This research was supported in part by Other Transaction award HR00112490375 from the U.S. Defense Advanced Research Projects Agency (DARPA) Friction for Accountability in Conversational Transactions (FACT) program.

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

# Appendix

## Table of Contents

## A    Experimental Setup

### A.1    Experimental Domain Details

Below we provide full details about the experimental domain we used in our experiments:

- **Square Assembly**: The goal is to pick up and place a square nut into the square peg. $\mathcal{D}_{\text{target}}$ consists of 10 demonstrations. $\mathcal{D}_{\text{prior}}$ consists of two different types of data: 200 episodes of *useful* demonstrations placing the nut into the goal peg, and 200 episodes of *adversarial* demonstrations that place the nut into the wrong peg. Note that the initial phase of the adversarial data also consists of useful motion for learning to pick up the nut. In our setup, the target task is in the same environment as the adversarial data while the useful data has a different background. We generated optimal demonstrations using scripted policies.

- **LIBERO-Can**: The goal is to pick up a can and place it into a basket. Corresponding task description is "pick up the alphabet soup and place it in the basket" and the environment is LIVING_ROOM_SCENE1 (from the LIBERO-90 suite). $\mathcal{D}_{\text{target}}$ consists of 10 human demonstrations from the LIBERO benchmark, and $\mathcal{D}_{\text{prior}}$ consists of 18 selected tasks from LIBERO-90 and each with 50 trajectories, resulting in a total of 900 trajectories. The list of task IDs in LIBERO-90 that was selected to form the prior dataset is:

```
"LIVING_ROOM_SCENE1_pick_up_the_ketchup_and_put_it_in_the_basket"
"LIVING_ROOM_SCENE2_pick_up_the_butter_and_put_it_in_the_basket"
"LIVING_ROOM_SCENE2_pick_up_the_orange_juice_and_put_it_in_the_basket"
"LIVING_ROOM_SCENE2_pick_up_the_tomato_sauce_and_put_it_in_the_basket"
"LIVING_ROOM_SCENE2_pick_up_the_milk_and_put_it_in_the_basket"
"LIVING_ROOM_SCENE2_pick_up_the_alphabet_soup_and_put_it_in_the_basket"
"LIVING_ROOM_SCENE3_pick_up_the_alphabet_soup_and_put_it_in_the_tray"
"LIVING_ROOM_SCENE3_pick_up_the_butter_and_put_it_in_the_tray"
"LIVING_ROOM_SCENE4_stack_the_left_bowl_on_the_right_bowl_and_place_them_in_the_tray"
"LIVING_ROOM_SCENE6_put_the_chocolate_pudding_to_the_left_of_the_plate"
"KITCHEN_SCENE1_open_the_bottom_drawer_of_the_cabinet"
"KITCHEN_SCENE2_put_the_middle_black_bowl_on_the_plate"
"KITCHEN_SCENE3_turn_on_the_stove_and_put_the_frying_pan_on_it"
"KITCHEN_SCENE8_turn_off_the_stove"
"KITCHEN_SCENE10_close_the_top_drawer_of_the_cabinet"
"STUDY_SCENE1_pick_up_the_book_and_place_it_in_the_front_compartment_of_the_caddy"
"STUDY_SCENE2_pick_up_the_book_and_place_it_in_the_back_compartment_of_the_caddy"
"STUDY_SCENE3_pick_up_the_white_mug_and_place_it_to_the_right_of_the_caddy"
```

- **Bridge-Pot** and **Bridge-Microwave**: In *Bridge-Pot* the robot picks up a pot on the burner and places into the sink. In *Bridge-Microwave*, the robot grasps the handle of the microwave door and

pulls it open. We leverage Bridge-V2 [7] as $\mathcal{D}_{\text{prior}}$ and collected $\mathcal{D}_{\text{target}}$ in our own setup with a ViperX arm [36], which is similar to some environments in Bridge-V2 but does not exist in the prior dataset. We use a ViperX arm [36] instead of the WidowX [37] as in Bridge-V2 dataset, which introduces additional domain difference for transferring useful pattern in prior data. We collected 10 demonstration for each of the target tasks using VR teleoperation and use a subset of Bridge-V2 as prior data. The full list of environments in the prior dataset is:

```
datacol1_toykitchen1
datacol1_toykitchen6
datacol2_folding_table
datacol2_robot_desk
datacol2_toykitchen1
datacol2_toykitchen5
datacol2_toykitchen7
datacol2_toysink2
deepthought_robot_desk
deepthought_toykitchen1
deepthought_toykitchen2
minsky_folding_table_white_tray
```

- **Franka-Pen-in-Cup**: The goal is to pick up a marker and put it into a cup. $\mathcal{D}_{\text{target}}$ consists of 10 human demonstrations. We tested two instances of $\mathcal{D}_{\text{prior}}$: 1) *PnP*: 105 trajectories of pick-and-place tasks from the same robot that we collected ourselves through VR teleoperation, and 2) *Wild*: 400 randomly sampled trajectories from the DROID [8] dataset (filtered to roughly match the viewpoint in target task). Pen-in-Cup is a task that exists in DROID but the demonstrations were collected in very different environments.

## A.2 Implementation Details of Baseline Methods and Ablations

We have two sets of implementations of FLOWRETRIEVAL and baselines for our experiments: 1) for experiments in simulation as well as the *Franka-Pen-in-Cup* task, we adapted the diffusion policy implementation of Chi et al. [33]; we use the U-Net variant of diffusion policy and load pretrained ImageNet weights for initializing the encoder network; 2) for Bridge-* tasks, we use the diffusion model implementation provided by Walke et al. [7][3]; we also load pretrained ImageNet weights for the encoder of the policy.

For **BR**, we re-implemented the pretraining logic for the VAE and reused the hyperparameters as documented in Du et al. [2]. For **SR** results in the paper, we re-implemented the pretraining logic for the latent skill space with the provided hyperparameters in Nasiriany et al. [1]. We also ran our simulation experiments using the full implementation of SAILOR provided by Nasiriany et al. [1], but fail to obtain non-zero success rates. One potential reason is that the tasks we consider in our experimental setup is relatively short, while SAILOR was designed for tackling long-horizon tasks (4-6 steps per task), such that our experimental domains may require a completely different set of hyperparameters than those used in the original work.

**ProprioRetrieval** is a method where we use proprioceptive data for computing similarity at the retrieval stage. We also apply flow loss during policy learning, and therefore it can be considered as an ablation of the retrieval space of FLOWRETRIEVAL. Specifically, we leverage end-effector Cartesian position difference between $s_t$ and $s_{t+k}$ to represent motion. While it is desirable to match the rotational motion as well, we find using the full end-effector Cartesian pose to compute similarity requires tuning scaling factors for balancing positional state versus rotational state (orientation) and not only does that require manual evaluations, but also existing datasets often use different conventions for orientation and therefore introduces additional challenges for comparing proprioceptive states across datasets directly. Hence in our implementation of **ProprioRetrieval**, we use only delta of the position of the end effector to retrieve from prior data. The feature $e_t$ for datapoint $s_t$ is computed as:

$$e_t = (s_t, s_{t+k}[: 3] - s_t[: 3]) \tag{7}$$

The similarity score is then computed as the negative $\ell_2$ distance between feature vectors.

## A.3 Retrieval Threshold Selection

In FLOWRETRIEVAL, we retrieve the top $\delta\%$ from the prior data. However, the optimal threshold can be different for different target task and prior dataset. Intuitively, if the similarity metric truly ranks the prior datapoints by usefulness to target task, we want to retrieve just the right amount of

---

[3] https://github.com/rail-berkeley/bridge_data_v2

| Method | SquareAssembly | LIBERO-Can | Bridge-Micro | Bridge-Pot | Pen-in-Cup |
|---|---|---|---|---|---|
| BC | 7% | 64% | 56% | 20% | 23% |
| BC-Co | 3% | 16% | 56% | 32% | 12% |
| FlowBC | 7% | 71% | 48% | 8% | 40% |
| ProrioRetrieval- | 44% | 73% | 24% | 40% | - |
| ProrioRetrieval | 54% | 81% | 16% | 12% | 44% |
| FLOWRETRIEVAL | **55%** | **90%** | **68%** | **64%** | **56%** |

Table 1: Success rates of baselines and ablations of FLOWRETRIEVAL.

data that supports the learning of target task. Du et al. [2] observed a bell-shaped curve between the relationship of retrieval threshold and policy performance. Due to constraints on computing resources, we only sparsely searched over a small range of threshold values for 3 tasks (2 in sim, 1 in real): Square Assembly, LIBERO-Can, and Bridge-Pot. In our experiments, we retrieve 35% from prior data in Square Assebmly, 10% in LIBERO-Can, and 1% in Bridge tasks. We then reuse the same threshold from bridge-Pot for the Bridge-Microwave task. For Franka Pen-in-Cup, we reuse the threshold of found in the Square Assembly task.

## B  Additional Baselines and Ablation Results

### B.1  Additional Baselines

We present the full evaluations of **BC-Co** and **FlowBC** in Table 1. **BC-Co** directly cotrains with the entire prior dataset. **FlowBC** applies auxiliary flow loss to **BC**. We see that while each could improve the performance from vanilla **BC** in certain domains, they can also hurt performance in others, depending on the composition of prior dataset.

### B.2  Using Proprioception for Retrieval

Low level motion features from proprioception are simple to compute and can be effective for extracting similar actions from past experience. Therefore we ablate our retrieval method with using propriceptive information, and denote this method as **ProprioRetrieval**. However, as proprioceptive features are entirely agnostic to the visual observation and therefore may retrieve irrelevant data such as those from very different viewpoints if the prior dataset has different viewpoints, e.g. OXE [13] and DROID [8].

**ProprioRetrieval** still applies the auxiliary flow loss during policy learning. We additionally evaluate a baseline version that does not use the flow loss as a comparison point, and use **ProprioRetrieval-** to denote this baseline. The success rates of both proprioception-based methods are reported in Table 1. We see that **ProprioRetrieval** can lead to high success rates when camera viewpoints are consistent (e.g. in Square Assembly and LIBERO-Can), but does not retrieve enough useful data from *Wild* to match the performance of FLOWRETRIEVAL in the Franka Pen-in-Cup task, and performs poorly in the Bridge tasks that have a large variety of prior tasks. Note that the downstream policy learning does not use the low-level action data to update the policy branck in the Franka Pen-in-Cup task when retrieving from the *Wild* dataset, due to the large variation in camera viewpoints. However, we still use the low level actions from the retrieved prior data of Bridge since the viewpoints are better aligned, but may introduce additional multimodality in policy learning.

# C  Retrieval Visualization

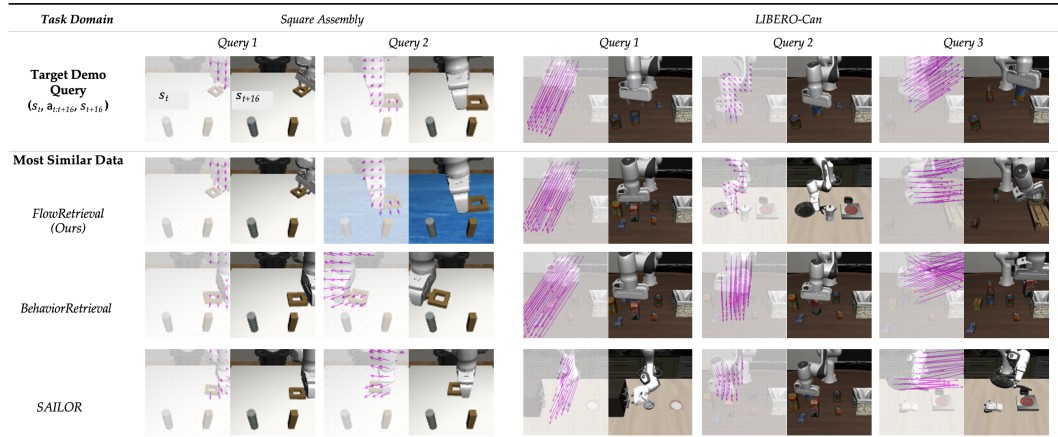

Figure 7: Visualization of paired retrieved data and query target datapoint by different methods in Square Assembly and LIBERO-Can.

Fig. 7 shows additional queries and retrieved data by different retrieval method in the two simulated tasks. BR and SR both encodes visual observations and low level actions in the latent space used for retrieval. We see that BR often overly focuses on visual scene similarity, and motion similarity is likely a second order feature (only effective if the visual scene is similar in the first place). Therefore in all cases, it cannot ignore the effect of the environment (background) and end up retrieving potentially adversarial data. SR can latch on either of the similarities, sometimes focusing on visual scene (Sqaure Assembly), and sometimes retrieving based on action similarity (LIBERO-Can). In contrast, FLOWRETRIEVAL consistently focuses on visual motion similarity.

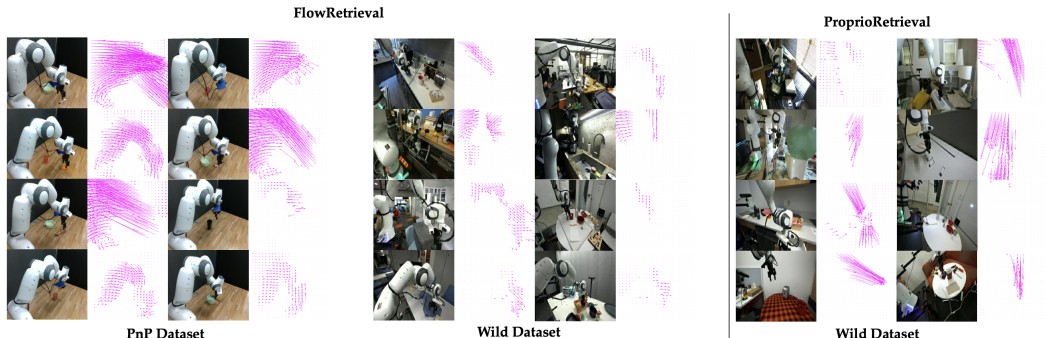

Figure 8: Visualization of example data points retrieved by FLOWRETRIEVAL and ProprioRetrieval in Franka Pen-in-Cup task.

Fig. 8 shows the example data points retrieved by FLOWRETRIEVAL and ProprioRetrieval in Franka Pen-in-Cup task. When retrieving from the *PnP* dataset, FLOWRETRIEVAL focuses on the pick-up and transfer stage of the target task, and does not retrieve the placing motions from the prior dataset, effectively filtering adversarial prior data. When retrieving from the *Wild* dataset, we see that FLOWRETRIEVAL retrieves viewpoints better aligned with that in the target task, while ProprioRetrieval retrieves very different viewpoints (sometimes the robot is not even in the view – see example in rightmost column, second from bottom). These data points are less informative for the downstream learning of the target task but may still provide additional regularization on learning a diverse set of visual features.

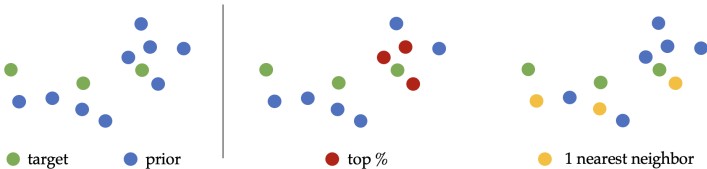

● target  ● prior  |  ● top %  |  ● 1 nearest neighbor

Figure 9: **Illustration of two retrieval strategies.** Depending on the distribution of target and prior data points, top-% may retrieve data only close to certain part of the target trajectory while KNN would retrieve uniformly.



Top 1%                    10-NN

Figure 10: Visualization of the retrieved datapoints in Bridge tasks using two different strategies.

## D  Ablating Retrieval Strategies

### D.1  KNN-based Retrieval Strategy

In our implementation of FLOWRETRIEVAL, we follow the practice of prior works and retrieve base on a threshold of the similarity score, taking the top $\delta\%$ closest datapoints from prior dataset to any point in target data. However, this does not augment the target task trajectory uniformly. As a result, when there are pauses in the dataset (which is true in most human-demonstrated datasets if not post-processed), we observe that the retrieved data contain a large portion of small motion data where the robot arm barely moves (Fig. 10 left).

One intuitive way to solve this issue is to retrieve top-$k$ data points for each state in the target task data. See an illustration of the two different retrieval strategies in Fig. 9. We visualize samples of the data retrieved by 10-NN in Bridge tasks in Fig. 10 (right) and see that it is able to retrieve meaningfully similar data to target task. However, we find that, when retrieving similar amount of total data as top 1%, this approach surprisingly does not lead to as performant policies as the existing approach, achieving 40% in Square Assembly (-15%), 80% in LIBERO-Can (-10%), and 60% (-8%) in Bridge-Microwave. One potential reason is that when we force each datapoint to have at least some datapoints retrieved from prior data, we might end up retrieving dissimilar and potentially adversarial data if the $k$ is not carefully selected for each datapoint.

### D.2  Retrieving with Pre-trained Representations

We evaluated retrieval in the Square Assembly task with pretrained visual representations to see if off-the-shelf models could be leveraged to retrieve motion-similar data. Specifically we take the delta between the features of $s_t$ and $s_{t+k}$ and use that to represent motion as proposed in prior work [38]. Fig. 11 shows the detailed analysis of data retrieved by Voltron [21], R3M [39], CLIP [40], and Dino-v2 [41]. We see that in general these models focus on visual features more than the motion itself and retrieving with such embeddings cannot bypass adversarial data in this task, with motion language-aligned models (Voltron and R3M) suffering the most.

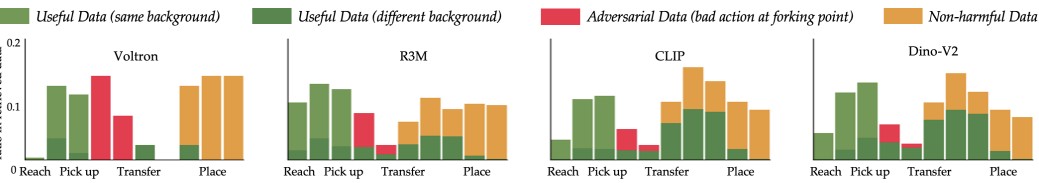

Figure 11: Retrieval Analysis for pretrained visual representations.

