# OpenReview forum: "FlowRetrieval: Flow-Guided Data Retrieval for Few-Shot Imitation Learning"
_robot-learning.org/CoRL/2024/Conference — CoRL 2024_

### Official Review · Reviewer_Z3WV · 2024-07-05
**use flow to mine useful prior data for few-shot manipulation learning**

**Originality:** 3
**Technical Quality:** 4
**Clarity Of Presentation:** 4
**Potential Impact:** 3
**Recommendation:** 3
**Confidence:** 4

**Review:**

(Strength)
1. The idea of finding data with similar motion patterns to augment the few-shot training set and therefore improve performance, is reasonable.
2. The proposed flow retrieval includes a VAE to conduct optical-flow-aware representation learning to support the retrieval and a policy learning that includes retrieved data and flow prediction as an auxiliary loss. This framework is a clean solution.
3. The paper is well-written and easy to follow.

(Weakness)
1. Lack of proper ablation studies to justify the design choice. The flow auxiliary loss branch is somewhat separated from the main flow retrieval idea. Please see more about this point in the question section. This is my major concern.

2. Need a sufficient justification of why flow-based retrieval is better than Proprio retrieval because it seems this paper assumes the target and prior datasets come from the same workspace. In this case, one would argue Proprio retrieval shall be better.

3. The figure 1 may be somewhat misleading. The co-training batch in Figure 1 includes data from two different environments. This implies that prior data and target data can come from drastically different sources. However, the experiments seem to focus on the setting where the prior and target come from a very similar setting.


Please see more details in the question section.

**Quality Of The Limitations Section:**

2

**Questions For Rebuttal:**

1. (Corresponds to W1) How much performance contribution comes from the flow auxiliary loss, instead of the proposed flow retrieval strategy?  If you haven't included this (in case I missed anything), I suggest "random retrieval + flow loss" as another comparison, instead of just FlowBC (flow loss without retrieval).

2. Is it possible to add some qualitative examples of why flow retrieval performs better than Proprio Retrieval?

3. Do you train one VAE for all environments, or each one separately?

4. Optical flow lacks real groundtruth and is usually very noisy when obtained from off-the-shelf models instead of simulation environments. I feel like the flow-aware representation learning part can be tricky, it would be better if more details are provided.

5. There are some recent works on one-shot imitation learning that share a very similar setting to this work, such as

     - One-shot Imitation Learning via Interaction Warping (CoRL2023)
     - One-Shot Imitation Learning- A Pose Estimation Perspective (CoRL2023)
     - One-Shot Imitation Learning with Invariance Matching for Robotic Manipulation (RSS2024)

   The authors are suggested to discuss them in the related work.

**Robotics Focus:**

4

**Summary Of Paper:**

The authors propose a few-shot manipulation learning method based on flow similarity. The proposed methods augment the small few-shot training set from a much larger prior dataset, by mining trajectories with similar motion patterns. The motion patterns are determined through representation learning with optical flow.

**Summary Of Recommendation:**

The flow retrieval few-shot learning method is sound but lacks some ablation studies to further justify the design choice. I list the current draft as "weak accept".

---

### Official Review · Reviewer_GJ3C · 2024-07-20
**Interesting and novel approach for few-shot skill imitation**

**Originality:** 4
**Technical Quality:** 4
**Clarity Of Presentation:** 4
**Potential Impact:** 3
**Recommendation:** 4
**Confidence:** 4

**Review:**

Strengths:

Novelty and Inspiration: While previous works have focused on affordance learning (often represented as optical flows) for manipulation skills, this is the first work that conduct skill/demonstration retrieval based on the motion similarity and is proved to be a better solution.

Comprehensive Experimentation: The experiments are thoroughly conducted, demonstrating comparisons with state-of-the-art demonstration retrieval based methods.

Weaknesses:

1) Dependence on Motion Similarity Representation: The overall performance of the method heavily relies on the motion similarity representation module. The policy learning module appears unable to correct errors introduced by inaccuracies in the motion similarity retrieval process.
2) Limited Skill Generation Capability: The motion similarity retrieval approach lacks dynamic programming, which limits its ability to stitch different skills together. As a result, the method's capacity to generate potential new skills from the dataset is constrained.

**Quality Of The Limitations Section:**

3

**Questions For Rebuttal:**

The paper is well-written, and the experiments are comprehensive, supporting the authors' proposal that motion similarity retrieval-based imitation is more beneficial for few-shot skill adaptation compared to visual similarity or language semantic similarity-based approaches.
Regarding the method and its potential, I have the following questions:

1) Motion Similarity and Affordance Learning:
The performance of motion similarity retrieval depends on the quality of the learned representations. Given that there are many affordance learning methods (including optical flow-based approaches) trained on large datasets, is there a way to leverage these existing methods to enhance the motion similarity retrieval process?

2) Dataset Size Impact:
How does the dataset size affect the imitation retrieval performance? It would be valuable to explore:
a) What is the minimum amount of training data required to learn a robust motion representation?
b) How does the dataset size correlate with the test performance? Is there a clear trend or diminishing returns past a certain point?

3) Cross-Embodiment Learning:
How does this method contribute to cross-embodiment learning scenarios? For instance:
a) If the robot executing the real task is significantly different from those in the dataset, can this method still effectively guide the unseen robot to accomplish the task?
b) What would be the expected performance of a novel robot with a different embodiment?
c) Are there any limitations or additional considerations when applying this method to robots with vastly different morphologies?

**Robotics Focus:**

4

**Summary Of Paper:**

This work proposes a novel retrieval-based few-shot imitation learning method that leverages motion similarities, specifically using optical flow between observation frames. The key insight of this research is that motion-based imitation retrieval offers a more robust approach compared to pure visual similarity methods, while providing more detailed information than high-level language-based semantic retrieval.

**Summary Of Recommendation:**

Recommendation: Accept with minor revisions. This paper presents a novel and promising approach to few-shot imitation learning using motion similarity-based retrieval. The work's key strength lies in its innovative use of optical flow to capture motion similarities, bridging the gap between visual and semantic retrieval methods. The comprehensive experiments demonstrate the potential advantages of this approach over existing visual and language-based methods.

---

### Official Review · Reviewer_JhhG · 2024-07-20

**Originality:** 4
**Technical Quality:** 4
**Clarity Of Presentation:** 4
**Potential Impact:** 3
**Recommendation:** 3
**Confidence:** 4

**Review:**

**Strengths**:
- The motivation behind the paper is strong. they have identified the major weaknesses of other retrieval methods and tried to address them by focusing on the motion similarity.
- the method is very simple and easy to understand / implement
- They have chosen their experiments to be very tailored to what their contributions are, which resulted in insightful conclusions.

**Weaknesses**:
- As the authors recognise themselves, this method requires to compare each frame of the target data with each frame of the prior data. This does not scale well with the size of either dataset.
- They briefly discuss that when using the Wild dataset they had to only train the flow prediction rather than the action prediction head. If I have understood this correctly then this is a big limitation as it means that even though something was retrieved because it was in the top d% of the prior dataset, it could still be overall harmful because the quality of the entire prior data is low.
    - Have you considered using a fixed threshold of similarity rather than retrieving the best d% ? This might mean that sometimes you retrieve less data, but if the data you are avoiding to retrieve was bad in the first place it should still help.

**Quality Of The Limitations Section:**

3

**Questions For Rebuttal:**

I would appreciate if the authors could comment on the few things I have mentioned above in the review.

A suggestion that I have is to add an interesting ablation to the Franka pen in cup experiment. More specifically, I would evaluate also FlowRetrieval with both PnP data and wild data. If your retrieval works well it should basically only retrieve PnP and therefore have a very similar performance to FlowRetrieval with only PnP.

**Robotics Focus:**

4

**Summary Of Paper:**

The authors propose a way of retrieving useful prior data when trying to complete a target task. They accomplish this by selecting the parts of the prior data which represent similar motions to the ones present in the target data. Additionally, in order to preserve the importance of motion during the learning and not only the retrieval, they include an auxiliary task of predicting the flow when training the policy. they finally provide a plethora of experiments both in simulation and real world scenarios.

**Summary Of Recommendation:**

The paper addresses the limitations of the previous work and is open about the limitations of the presented work. They provide many experiments that are tailored to their work and therefore prove to be interesting.

---

### Decision · Program_Chairs · 2024-09-04

**Decision:**

Accept

**Comment:**

The paper introduces a new method for few-shot imitation learning in robot manipulation. The main idea is to use optical flow estimation from the few-shot demonstration videos to retrieve similar data from prior datasets to improve the policy learning.

Strengths

- The reviewers agrees on the novelty of the proposed method by using optical flow for data retrieval.

- The paper is well-written with solid experiments to demonstrate the effectiveness of the proposed method.

Weaknesses

The reviewers have also raised several concerns of the paper.

- Scalability of the proposed method on the size of the dataset.
- The effect of the quality of the retrieved data on the policy.
- Missing citations and some ablation study of the proposed method.

Post-Rebuttal
- The authors have successfully addressed the concerns from the reviewers.